# ES-MAML: Simple Hessian-Free Meta Learning

**Xingyou Song**[*]**, Yuxiang Yang**[‡]**, Krzysztof Choromanski**
Google Brain
{xingyousong,yxyang,kchoro}@google.com

**Aldo Pacchiano**
UC Berkeley
pacchiano@berkeley.edu

**Wenbo Gao**[*†]**, Yunhao Tang**[†]
Columbia University
{wg2279,yt2541}@columbia.edu

## ABSTRACT

We introduce *ES-MAML*, a new framework for solving the *model agnostic meta learning* (MAML) problem based on *Evolution Strategies* (ES). Existing algorithms for MAML are based on policy gradients, and incur significant difficulties when attempting to estimate second derivatives using backpropagation on stochastic policies. We show how ES can be applied to MAML to obtain an algorithm which avoids the problem of estimating second derivatives, and is also conceptually simple and easy to implement. Moreover, ES-MAML can handle new types of non-smooth adaptation operators, and other techniques for improving performance and estimation of ES methods become applicable. We show empirically that ES-MAML is competitive with existing methods and often yields better adaptation with fewer queries.

## 1 INTRODUCTION

*Meta-learning* is a paradigm in machine learning that aims to develop models and training algorithms which can quickly adapt to new tasks and data. Our focus in this paper is on meta-learning in reinforcement learning (RL), where data efficiency is of paramount importance because gathering new samples often requires costly simulations or interactions with the real world. A popular technique for RL meta-learning is *Model Agnostic Meta Learning* (MAML) (Finn et al., 2017; 2018), a model for training an agent which can quickly adapt to new and unknown tasks by performing one (or a few) gradient updates in the new environment. We provide a formal description of MAML in Section 2.

MAML has proven to be successful for many applications. However, implementing and running MAML continues to be challenging. One major complication is that the standard version of MAML requires estimating second derivatives of the RL reward function, which is difficult when using backpropagation on stochastic policies; indeed, the original implementation of MAML (Finn et al., 2017) did so incorrectly, which spurred the development of unbiased higher-order estimators (DiCE, (Foerster et al., 2018)) and further analysis of the credit assignment mechanism in MAML (Rothfuss et al., 2019). Another challenge arises from the high variance inherent in policy gradient methods, which can be ameliorated through control variates such as in T-MAML (Liu et al., 2019), through careful adaptive hyperparameter tuning (Behl et al., 2019; Antoniou et al., 2019) and learning rate annealing (Loshchilov & Hutter, 2017).

To avoid these issues, we propose an alternative approach to MAML based on *Evolution Strategies* (ES), as opposed to the policy gradient underlying previous MAML algorithms. We provide a detailed discussion of ES in Section 3.1. ES has several advantages:

---

[*]Equal contribution.
[†]Work performed during Google internship.
[‡]Work performed during the Google AI Residency Program. http://g.co/airesidency

1. Our zero-order formulation of ES-MAML (Section 3.2, Algorithm 3) does not require estimating any second derivatives. This dodges the many issues caused by estimating second derivatives with backpropagation on stochastic policies (see Section 2 for details).

2. ES is conceptually much simpler than policy gradients, which also translates to ease of implementation. It does not use backpropagation, so it can be run on CPUs only.

3. ES is highly flexible with different adaptation operators (Section 3.3).

4. ES allows us to use deterministic policies, which can be safer when doing adaptation (Section 4.3). ES is also capable of learning linear and other compact policies (Section 4.2).

On the point (4), a feature of ES algorithms is that exploration takes place in the parameter space. Whereas policy gradient methods are primarily motivated by interactions with the environment through randomized actions, ES is driven by optimization in high-dimensional parameter spaces with an expensive querying model. In the context of MAML, the notions of "exploration" and "task identification" have thus been shifted to the parameter space instead of the action space. This distinction plays a key role in the stability of the algorithm. One immediate implication is that we can use deterministic policies, unlike policy gradients which is based on stochastic policies. Another difference is that ES uses only the total reward and not the individual state-action pairs within each episode. While this may appear to be a weakness, since less information is being used, we find in practice that it seems to lead to more stable training profiles.

This paper is organized as follows. In Section 2, we give a formal definition of MAML, and discuss related works. In Section 3, we introduce Evolutionary Strategies and show how ES can be applied to create a new framework for MAML. In Section 4, we present numerical experiments, highlighting the topics of exploration (Section 4.1), the utility of compact architectures (Section 4.2), the stability of deterministic policies (Section 4.3), and comparisons against existing MAML algorithms in the few-shot regime (Section 4.4). Additional material can be found in the Appendix.

## 2    MODEL AGNOSTIC META LEARNING IN RL

We first discuss the original formulation of MAML (Finn et al., 2017). Let $\mathcal{T}$ be a set of reinforcement learning tasks with common state and action spaces $\mathcal{S}, \mathcal{A}$, and $\mathcal{P}(\mathcal{T})$ a distribution over $\mathcal{T}$. In the standard MAML setting, each task $T_i \in \mathcal{T}$ has an associated Markov Decision Process (MDP) with transition distribution $q_i(s_{t+1}|s_t, a_t)$, an episode length $H$, and a reward function $R_{T_i}$ which maps a trajectory $\tau = (s_0, a_1, ..., a_{H-1}, s_H)$ to the total reward $R(\tau)$. A *stochastic policy* is a function $\pi : \mathcal{S} \to \mathcal{P}(\mathcal{A})$ which maps states to probability distributions over the action space. A *deterministic policy* is a function $\pi : \mathcal{S} \to \mathcal{A}$. Policies are typically encoded by a neural network with parameters $\theta$, and we often refer to the policy $\pi_\theta$ simply by $\theta$.

The MAML problem is to find the so-called *MAML point* (called also a *meta-policy*), which is a policy $\theta^*$ that can be 'adapted' quickly to solve an unknown task $T \in \mathcal{T}$ by taking a (few)[1] policy gradient steps with respect to $T$. The optimization problem to be solved in training (in its one-shot version) is thus of the form:

$$\max_\theta J(\theta) := \mathbb{E}_{T \sim \mathcal{P}(\mathcal{T})}[\mathbb{E}_{\tau' \sim \mathcal{P}_T(\tau'|\theta')}[R_T(\tau')]], \tag{1}$$

where: $\theta' = U(\theta, T) = \theta + \alpha \nabla_\theta \mathbb{E}_{\tau \sim \mathcal{P}_T(\tau|\theta)}[R_T(\tau)]$ is called the *adapted policy* for a step size $\alpha > 0$ and $\mathcal{P}_T(\cdot|\eta)$ is a distribution over trajectories given task $T \in \mathcal{T}$ and conditioned on the policy parameterized by $\eta$.

Standard MAML approaches are based on the following expression for the gradient of the MAML objective function (1) to conduct training:

$$\nabla_\theta J(\theta) = \mathbb{E}_{T \sim \mathcal{P}(\mathcal{T})}[\mathbb{E}_{\tau' \sim \mathcal{P}_T(\tau'|\theta')}[\nabla_{\theta'} \log \mathcal{P}_T(\tau'|\theta') R_T(\tau') \nabla_\theta U(\theta, T)]]. \tag{2}$$

We collectively refer to algorithms based on computing (2) using policy gradients as *PG-MAML*.

---

[1]We adopt the common convention of defining the adaptation operator with a single gradient step, to simplify notation. It can be extended to multiple steps.

Since the adaptation operator $U(\theta, T)$ contains the policy gradient $\nabla_\theta \mathbb{E}_{\tau \sim \mathcal{P}_T(\tau|\theta)}[R(\tau)]$, its own gradient $\nabla_\theta U(\theta, T)$ is second-order in $\theta$:

$$\nabla_\theta U = \mathbf{I} + \alpha \int \mathcal{P}_T(\tau|\theta) \nabla_\theta^2 \log \pi_\theta(\tau) R_T(\tau) d\tau + \alpha \int \mathcal{P}_T(\tau|\theta) \nabla_\theta \log \pi_\theta(\tau) \nabla_\theta \log \pi_\theta(\tau)^T R_T(\tau) d\tau. \tag{3}$$

Correctly computing the gradient (2) with the term (3) using automatic differentiation is known to be tricky. Multiple authors (Foerster et al., 2018; Rothfuss et al., 2019; Liu et al., 2019) have pointed out that the original implementation of MAML incorrectly estimates the term (3), which inadvertently causes the training to lose 'pre-adaptation credit assignment'. Moreover, even when correctly implemented, the variance when estimating (3) can be extremely high, which impedes training. To improve on this, extensions to the original MAML include ProMP (Rothfuss et al., 2019), which introduces a new low-variance curvature (LVC) estimator for the Hessian, and T-MAML (Liu et al., 2019), which adds control variates to reduce the variance of the unbiased DiCE estimator (Foerster et al., 2018). However, these are not without their drawbacks: the proposed solutions are complicated, the variance of the Hessian estimate remains problematic, and LVC introduces unknown estimator bias.

Another issue that arises in PG-MAML is that policies are necessarily stochastic. However, randomized actions can lead to risky exploration behavior when computing the adaptation, especially for robotics applications where the collection of tasks may involve differing system dynamics as opposed to only differing rewards (Yang et al., 2019). We explore this further in Section 4.3.

These issues: the difficulty of estimating the Hessian term (3), the typically high variance of $\nabla_\theta J(\theta)$ for policy gradient algorithms in general, and the unsuitability of stochastic policies in some domains, lead us to the proposed method ES-MAML in Section 3.

Aside from policy gradients, there have also been biologically-inspired algorithms for MAML, based on concepts such as the Baldwin effect (Fernando et al., 2018). However, we note that despite the similar naming, methods such as 'Evolvability ES' (Gajewski et al., 2019) bear little resemblance to our proposed ES-MAML. The problem solved by our algorithm is the standard MAML, whereas (Gajewski et al., 2019) aims to maximize loosely related notions of the *diversity of behavioral characteristics*. Moreover, ES-MAML and its extensions we consider are all derived notions such as smoothings and approximations, with rigorous mathematical definitions as stated below.

## 3    ES-MAML ALGORITHMS

Formulating MAML with ES allows us to employ numerous techniques originally developed for enhancing ES, to MAML. We aim to improve both phases of MAML algorithm: the meta-learning training algorithm, and the efficiency of the adaptation operator.

### 3.1    EVOLUTION STRATEGIES METHODS (ES)

Evolution Strategies (ES) (Wierstra et al., 2008; 2014), which recently became popular for RL (Salimans et al., 2017), rely on optimizing the smoothing of the blackbox function $f : \mathbb{R}^d \to \mathbb{R}$, which takes as input parameters $\theta \in \mathbb{R}^d$ of the policy and outputs total discounted (expected) reward obtained by an agent applying that policy in the given environment. Instead of optimizing the function $f$ directly, we optimize a smoothed objective. We define the Gaussian smoothing of $F$ as $\tilde{f}_\sigma(\theta) = \mathbb{E}_{\mathbf{g} \sim \mathcal{N}(0, \mathbb{I}_d)}[f(\theta + \sigma \mathbf{g})]$. The gradient of this smoothed objective, sometimes called an *ES-gradient*, is given as (see: (Nesterov & Spokoiny, 2017)):

$$\nabla_\theta \tilde{f}_\sigma(\theta) = \frac{1}{\sigma} \mathbb{E}_{\mathbf{g} \sim \mathcal{N}(0, \mathbf{I}_d)}[f(\theta + \sigma \mathbf{g})\mathbf{g}]. \tag{4}$$

Note that the gradient can be approximated via Monte Carlo (MC) samples:

In ES literature the above algorithm is often modified by adding control variates to equation 4 to obtain other unbiased estimators with reduced variance. The *forward finite difference (Forward-FD)* estimator (Choromanski et al., 2018) is given by subtracting the current policy value $f(\theta)$, yielding $\nabla_\theta \tilde{f}_\sigma(\theta) = \frac{1}{\sigma} \mathbb{E}_{\mathbf{g} \sim \mathcal{N}(0, \mathbf{I}_d)}[(f(\theta + \sigma \mathbf{g}) - f(\theta))\mathbf{g}]$. The *antithetic* estimator (Nesterov & Spokoiny, 2017; Mania et al., 2018) is given by the symmetric difference $\nabla_\theta \tilde{f}_\sigma(\theta) = \frac{1}{2\sigma} \mathbb{E}_{\mathbf{g} \sim \mathcal{N}(0, \mathbf{I}_d)}[(f(\theta +$

**1 ESGrad** $(f, \theta, n, \sigma)$
**inputs:** function $f$, policy $\theta$, number of perturbations $n$, precision $\sigma$
**2** Sample $n$ i.i.d $N(0, I)$ vectors $g_1, \ldots, g_n$;
**3** **return** $\frac{1}{n\sigma} \sum_{i=1}^{n} f(\theta + \sigma g_i) g_i$;

**Algorithm 1:** Monte Carlo ES Gradient

$\sigma \mathbf{g}) - f(\theta - \sigma \mathbf{g}))\mathbf{g}]$. Notice that the variance of the Forward-FD and antithetic estimators is translation-invariant with respect to $f$. In practice, the Forward-FD or antithetic estimator is usually preferred over the basic version expressed in equation 4.

In the next sections we will refer to Algorithm 1 for computing the gradient though we emphasize that there are several other recently developed variants of computing ES-gradients as well as applying them for optimization. We describe some of these variants in Section 3.3 and appendix A.3. A key feature of ES-MAML is that we can directly make use of new enhancements of ES.

## 3.2 META-TRAINING MAML WITH ES

To formulate MAML in the ES framework, we take a more abstract viewpoint. For each task $T \in \mathcal{T}$, let $f^T(\theta)$ be the (expected) cumulative reward of the policy $\theta$. We treat $f^T$ as a blackbox, and make no assumptions on its structure (so the task need not even be MDP, and $f^T$ may be nonsmooth). The MAML problem is then

$$\max_{\theta} J(\theta) := \mathbb{E}_{T \sim \mathcal{P}(\mathcal{T})} f^T(U(\theta, T)). \tag{5}$$

As argued in (Liu et al., 2019; Rothfuss et al., 2019) (see also Section 2), a major challenge for policy gradient MAML is estimating the Hessian, which is both conceptually subtle and difficult to correctly implement using automatic differentiation. The algorithm we propose obviates the need to calculate any second derivatives, and thus avoids this issue.

Suppose that we can evaluate (or approximate) $f^T(\theta)$ and $U(\theta, T)$, but $f^T$ and $U(\cdot, T)$ may be nonsmooth or their gradients may be intractable. We consider the Gaussian smoothing $\widetilde{J}_\sigma$ of the MAML reward (5), and optimize $\widetilde{J}_\sigma$ using ES methods. The gradient $\nabla \widetilde{J}_\sigma(\theta)$ is given by

$$\nabla \widetilde{J}_\sigma(\theta) = \mathbb{E}_{\substack{T \sim \mathcal{P}(\mathcal{T}) \\ \mathbf{g} \sim \mathcal{N}(0, \mathbf{I})}} \left[ \frac{1}{\sigma} f^T(U(\theta + \sigma \mathbf{g}, T)) \mathbf{g} \right] \tag{6}$$

and can be estimated by jointly sampling over $(T, \mathbf{g})$ and evaluating $f^T(U(\theta + \sigma \mathbf{g}, T))$. This algorithm is specified in Algorithm 2 box, and we refer to it as *(zero-order) ES-MAML*.

**Data:** initial policy $\theta_0$, meta step size $\beta$
**1 for** $t = 0, 1, \ldots$ **do**
**2** Sample $n$ tasks $T_1, \ldots, T_n$ and iid vectors $\mathbf{g}_1, \ldots, \mathbf{g}_n \sim \mathcal{N}(0, \mathbf{I})$;
**3** **foreach** $(T_i, \mathbf{g}_i)$ **do**
**4** $\quad | \quad v_i \leftarrow f^{T_i}(U(\theta_t + \sigma \mathbf{g}_i, T_i))$
**5** **end**
**6** $\theta_{t+1} \leftarrow \theta_t + \frac{\beta}{\sigma n} \sum_{i=1}^{n} v_i \mathbf{g}_i$
**7 end**

**Algorithm 2:** Zero-Order ES-MAML (general adaptation operator $U(\cdot, T)$)

**Data:** initial policy $\theta_0$, adaptation step size $\alpha$, meta step size $\beta$, number of queries $K$
**1 for** $t = 0, 1, \ldots$ **do**
**2** Sample $n$ tasks $T_1, \ldots, T_n$ and iid vectors $\mathbf{g}_1, \ldots, \mathbf{g}_n \sim \mathcal{N}(0, \mathbf{I})$;
**3** **foreach** $(T_i, \mathbf{g}_i)$ **do**
**4** $\quad | \quad \mathbf{d}^{(i)} \leftarrow \text{ESGRAD}(f^{T_i}, \theta_t + \sigma \mathbf{g}_i, K, \sigma)$;
**5** $\quad | \quad \theta_t^{(i)} \leftarrow \theta_t + \sigma \mathbf{g}_i + \alpha \mathbf{d}^{(i)}$;
**6** $\quad | \quad v_i \leftarrow f^{T_i}(\theta_t^{(i)})$;
**7** **end**
**8** $\theta_{t+1} \leftarrow \theta_t + \frac{\beta}{\sigma n} \sum_{i=1}^{n} v_i \mathbf{g}_i$;
**9 end**

**Algorithm 3:** Zero-Order ES-MAML with ES-Gradient Adaptation

The standard adaptation operator $U(\cdot, T)$ is the one-step task gradient. Since $f^T$ is permitted to be nonsmooth in our setting, we use the adaptation operator $U(\theta, T) = \theta + \alpha \nabla \widetilde{f}_\sigma^T(\theta)$ acting on its smoothing. Expanding the definition of $\widetilde{J}_\sigma$, the gradient of the smoothed MAML is then given by

$$\nabla \widetilde{J}_\sigma(\theta) = \frac{1}{\sigma} \mathbb{E}_{\substack{T \sim \mathcal{P}(\mathcal{T}) \\ \mathbf{g} \sim \mathcal{N}(0, \mathbf{I})}} \left[ f^T \left( \theta + \sigma \mathbf{g} + \frac{1}{\sigma} \mathbb{E}_{\mathbf{h} \sim \mathcal{N}(0, \mathbf{I})} [f^T(\theta + \sigma \mathbf{g} + \sigma \mathbf{h}) \mathbf{h}] \right) \mathbf{g} \right]. \tag{7}$$

This leads to the algorithm that we specify in Algorithm 3, where the adaptation operator $U(\cdot, T)$ is itself estimated using the ES gradient in the inner loop.

We can also derive an algorithm analogous to PG-MAML by applying a first-order method to the MAML reward $\mathbb{E}_{T \sim \mathcal{P}(\mathcal{T})} \widetilde{f}^T(\theta + \alpha \nabla \widetilde{f}^T(\theta))$ directly, without smoothing. The gradient is given by

$$\nabla J(\theta) = \mathbb{E}_{T \sim \mathcal{P}(\mathcal{T})} \nabla \widetilde{f}^T(\theta + \alpha \nabla \widetilde{f}^T(\theta))(\mathbf{I} + \alpha \nabla^2 \widetilde{f}^T(\theta)), \tag{8}$$

which corresponds to equation (3) in (Liu et al., 2019) when expressed in terms of policy gradients. Every term in this expression has a simple Monte Carlo estimator (see Algorithm 4 in the appendix for the MC Hessian estimator). We discuss this algorithm in greater detail in Appendix A.1. This formulation can be viewed as the "**MAML of the smoothing**", compared to the "**smoothing of the MAML**" which is the basis for Algorithm 3. It is the additional smoothing present in equation 6 which eliminates the gradient of $U(\cdot, T)$ (and hence, the Hessian of $f^T$). Just as with the Hessian estimation in the original PG-MAML, we find empirically that the MC estimator of the Hessian (Algorithm 4) has high variance, making it often harmful in training. We present some comparisons between Algorithm 3 and Algorithm 5, with and without the Hessian term, in Appendix A.1.2.

Note that when $U(\cdot, T)$ is estimated, such as in Algorithm 3, the resulting estimator for $\nabla \widetilde{J}_\sigma$ will in general be biased. This is similar to the estimator bias which occurs in PG-MAML because we do not have access to the true adapted trajectory distribution. We discuss this further in Appendix A.2.

### 3.3 IMPROVING THE ADAPTATION OPERATOR WITH ES

Algorithm 2 allows for great flexibility in choosing new adaptation operators. The simplest extension is to modify the ES gradient step: we can draw on general techniques for improving the ES gradient estimator, some of which are described in Appendix A.3. Some other methods are explored below.

#### 3.3.1 IMPROVED EXPLORATION

Instead of using i.i.d Gaussian vectors to estimate the ES gradient in $U(\cdot, T)$, we consider samples constructed according to *Determinantal Point Processes* (DPP). DPP sampling (Kulesza & Taskar, 2012; Wachinger & Golland, 2015) is a method of selecting a subset of samples so as to maximize the 'diversity' of the subset. It has been applied to ES to select perturbations $\mathbf{g}_i$ so that the gradient estimator has lower variance (Choromanski et al., 2019a). The sampling matrix determining DPP sampling can also be *data-dependent* and use information from the meta-training stage to construct a learned kernel with better properties for the adaptation phase. In the experimental section we show that DPP-ES can help in improving adaptation in MAML.

#### 3.3.2 HILL CLIMBING AND POPULATION SEARCH

Nondifferentiable operators $U(\cdot, T)$ can be also used in Algorithm 2. One particularly interesting example is the *local search* operator given by $U(\theta, T) = \mathrm{argmax}\{f^T(\theta') : \|\theta' - \theta\| \leq R\}$, where $R > 0$ is the search radius. That is, $U(\theta, T)$ selects the best policy for task $T$ which is in a 'neighborhood' of $\theta$. For simplicity, we took the search neighborhood to be the ball $B(\theta, R)$ here, but we may also use more general neighborhoods of $\theta$. In general, exactly solving for the maximizer of $f^T$ over $B(\theta, R)$ is intractable, but local search can often be well approximated by a *hill climbing* algorithm. Hill climbing creates a population of candidate policies by perturbing the best observed policy (which is initialized to $\theta$), evaluates the reward $f^T$ for each candidate, and then updates the best observed policy. This is repeated for several iterations. A key property of this search method is that the progress is *monotonic*, so the reward of the returned policy $U(\theta, T)$ will always improve over $\theta$. This does not hold for the stochastic gradient operator, and appears to be beneficial on some difficult problems (see Section 4.1). It has been claimed that hill climbing and other genetic algorithms (Moriarty et al., 1999) are competitive with gradient-based methods for solving difficult RL tasks (Such et al., 2017; Risi & Stanley, 2019). Another stochastic algorithm approximating local search is *CMA-ES* (Hansen et al., 2003; Igel, 2003; Krause et al., 2016), which performs more sophisticated search by adapting the covariance matrix of the perturbations.

Figure 1: (a) ES-MAML and PG-MAML exploration behavior. (b) Different exploration methods when $K$ is limited ($K = 5$ plotted with lighter colors) or large penalties are added on wrong goals.

(a)                                                          (b)

## 4 EXPERIMENTS

The performance of MAML algorithms can be evaluated in several ways. One important measure is the performance of the final meta-policy: whether the algorithm can consistently produce meta-policies with better adaptation. In the RL setting, the adaptation of the meta-policy is also a function of the number $K$ of *queries* used: that is, the number of rollouts used by the adaptation operator $U(\cdot, T)$. The meta-learning goal of data efficiency corresponds to adapting with low $K$. The speed of the meta-training is also important, and can be measured in several ways: the number of meta-policy updates, wall-clock time, and the number of rollouts used for meta-training. In this section, we present experiments which evaluate various aspects of ES-MAML and PG-MAML in terms of data efficiency ($K$) and meta-training time. Further details of the environments and hyperparameters are given in Appendix A.7.

In the RL setting, the amount of information used drastically decreases if ES methods are applied in comparison to the PG setting. To be precise, ES uses only the cumulative reward over an episode, whereas policy gradients use every state-action pair. Intuitively, we may thus expect that ES should have worse sampling complexity because it uses less information for the same number of rollouts. However, it seems that in practice ES often matches or even exceeds policy gradients approaches (Salimans et al., 2017; Mania et al., 2018). Several explanations have been proposed: In the PG case, especially with algorithms such as PPO, the network must optimize multiple additional surrogate objectives such as entropy bonuses and value functions as well as hyperparameters such as the TD-step number. Furthermore, it has been argued that ES is more robust against delayed rewards, action infrequency, and long time horizons (Salimans et al., 2017). These advantages of ES in traditional RL also transfer to MAML, as we show empirically in this section. ES may lead to additional advantages (even if the numbers of rollouts needed in training is comparable with PG ones) in terms of wall-clock time, because it does not require backpropagation, and can be parallelized over CPUs.

### 4.1 EXPLORATION: TARGET ENVIRONMENTS

In this section, we present two experiments on environments with very sparse rewards where the meta-policy must exhibit *exploratory* behavior to determine the correct adaptation.

The *four corners* benchmark was introduced in (Rothfuss et al., 2019) to demonstrate the weaknesses of exploration in PG-MAML. An agent on a 2D square receives reward for moving towards a selected corner of the square, but only observes rewards once it is sufficiently close to the target corner, making the reward sparse. An effective exploration strategy for this set of tasks is for the meta-policy $\theta^*$ to travel in circular trajectories to observe which corner produces rewards; however, for a single policy to produce this exploration behavior is difficult. In Figure 1, we demonstrate the behavior of ES-MAML on the four corners problem. When $K = 20$, the same number of rollouts for adaptation as used in (Rothfuss et al., 2019), the basic version of Algorithm 3 is able to correctly explore and adapt to the task by finding the target corner. Moreover, it does not require any modifications to encourage exploration, unlike PG-MAML. We further used $K = 10, 5$, which caused the performance to drop. For better performance in this low-information environment, we experimented with two different adaptation operators $U(\cdot, T)$ in Algorithm 2, which are HC (hill climbing) and DPP-ES. The standard ES gradient is denoted MC.

Furthermore, ES-MAML is not limited to "single goal" exploration. We created a more difficult task, *six circles*, where the agent continuously accrues negative rewards until it reaches six target points to "deactivate" them. Solving this task requires the agent to explore in circular trajectories, similar to the trajectory used by PG-MAML on the four corners task. We visualize the behavior in Figure 2. Observe that ES-MAML with the HC operator is able to develop a strategy to explore the target locations.

Figure 2: ES-MAML exploration on six circle task ($K = 20$).

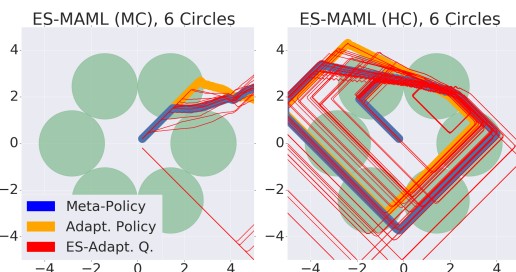

From Figure 1, we observed that both operators DPP-ES and HC were able to improve exploration performance. We also created a modified task by heavily penalizing incorrect goals, which caused performance to dramatically drop for MC and DPP-ES. This is due to the variance from the MC-gradient, which may result in an adapted policy that accidentally produces large negative rewards or become stuck in local-optima (i.e. refuse to explore due to negative rewards). This is also fixed by the HC adaptation, which enforces non-decreasing rewards during adaptation, allowing the ES-MAML to progress.

Additional examples on the classic Navigation-2D task are presented in Appendix A.4, highlighting the differences in exploration behavior between PG-MAML and ES-MAML.

## 4.2 GOOD ADAPTATION WITH COMPACT ARCHITECTURES

One of the main benefits of ES is due to its ability to train compact linear policies, which can outperform hidden-layer policies. We demonstrate this on several benchmark MAML problems in the HalfCheetah and Ant environments in Figure 3. In contrast, (Finn & Levine, 2018) observed that PG-MAML empirically and theoretically suggested that training with more deeper layers under SGD increases performance. We demonstrate that on the Forward-Backward and Goal-Velocity MAML benchmarks, ES-MAML is consistently able to train successful linear policies faster than deep networks. We also show that, for the Forward-Backward Ant problem, ES-MAML with the new HC operator is the most performant. Using more compact policies also directly speeds up ES-MAML, since fewer perturbations are needed for gradient estimation.

Figure 3: The Forward-Backward and Goal-Velocity MAML problems. We compare the performance for Linear (L) policies and policies with one hidden layer (H) for different $K$.

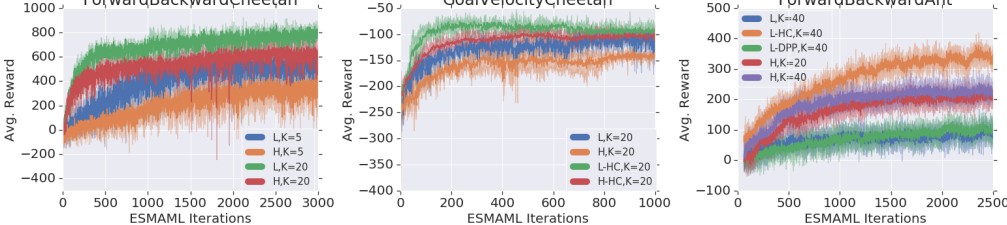

## 4.3 DETERMINISTIC POLICIES

We find that deterministic policies often produce more stable behaviors than the stochastic ones that are required for PG, where randomized actions in unstable environments can lead to catastrophic outcomes. In PG, this is often mitigated by reducing the entropy bonus, but this has an undesirable side effect of reducing exploration. In contrast, ES-MAML explores in parameter space, which mitigates this issue. To demonstrate this, we use the "Biased-Sensor CartPole" environment from (Yang et al., 2019). This environment has unstable dynamics and sparse rewards, so it requires exploration but is also risky. We see in Figure 4 that ES-MAML is able to stably maintain the maximum reward (500).

Figure 4: Stability comparisons of ES and PG on the Biased-Sensor CartPole and Swimmer, Walker2d environments. (L), (H), and (HH) denote linear, one- and two-hidden layer policies.

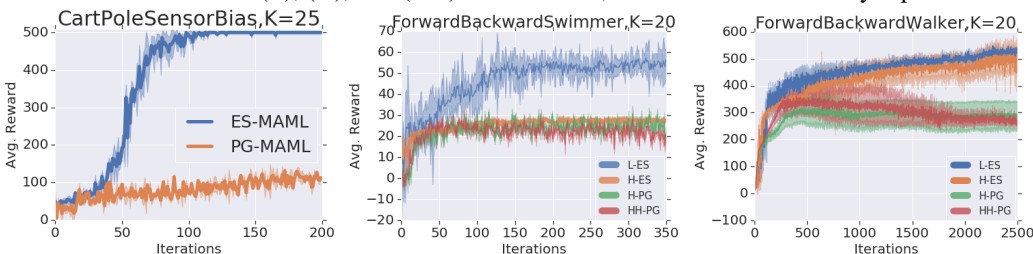

We also include results in Figure 4 from two other environments, Swimmer and Walker2d, for which it is known that PG is surprisingly unstable, and ES yields better training (Mania et al., 2018). Notice that we again find linear policies (L) outperforming policies with one (H) or two (HH) hidden layers.

## 4.4 LOW-$K$ BENCHMARKS

For real-world applications, we may be constrained to use fewer queries $K$ than has typically been demonstrated in previous MAML works. Hence, it is of interest to compare how ES-MAML compares to PG-MAML for adapting with very low $K$.

One possible concern is that low $K$ might harm ES in particular because it uses only the cumulative rewards; if for example $K = 5$, then the ES adaptation gradient can make use of only 5 values. In comparison, PG-MAML uses $K \cdot H$ state-action pairs, so for $K = 5, H = 200$, PG-MAML still has 1000 pieces of information available.

However, we find experimentally that the standard ES-MAML (Algorithm 3) remains competitive with PG-MAML even in the low-$K$ setting. In Figure 5, we compare ES-MAML and PG-MAML on the Forward-Backward and Goal-Velocity tasks across four environments (HalfCheetah, Swimmer, Walker2d, Ant) and two model architectures. While PG-MAML can generally outperform ES-MAML on the Goal-Velocity task, ES-MAML is similar or better on the Forward-Backward task. Moreover, we observed that for low $K$, PG-MAML can be highly unstable (note the wide error bars), with some trajectories failing catastrophically, whereas ES-MAML is relatively stable. This is an important consideration in real applications, where the risk of catastrophic failure is undesirable.

Figure 5: Low $K$ comparisons between ES-MAML and PG-MAML.

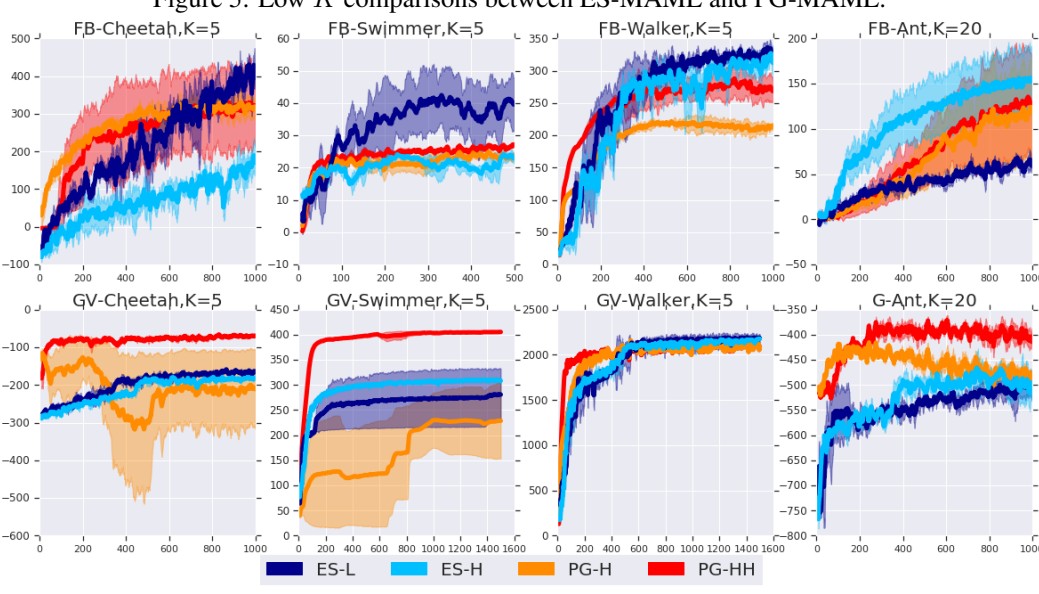

## 5 CONCLUSION

We have presented a new framework for MAML based on ES algorithms. The ES-MAML approach avoids the problems of Hessian estimation which necessitated complicated alterations in PG-MAML and is straightforward to implement. ES-MAML is flexible in the choice of adaptation operators, and can be augmented with general improvements to ES, along with more exotic adaptation operators. In particular, ES-MAML can be paired with nonsmooth adaptation operators such as hill climbing, which we found empirically to yield better exploratory behavior and better performance on sparse-reward environments. ES-MAML performs well with linear or compact *deterministic* policies, which is an advantage when adapting if the state dynamics are possibly unstable.

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

## A.1 FIRST-ORDER ES-MAML

### A.1.1 ALGORITHM

Suppose that we *first* apply Gaussian smoothing to the task rewards and then form the MAML problem, so we have $J(\theta) = \mathbb{E}_{T \sim \mathcal{P}(\mathcal{T})} \widetilde{f}^T(U(\theta, T))$. The function $J$ is then itself differentiable, and we can directly apply first-order methods to it. The classical case where $U(\theta, T) = \theta + \alpha \nabla \widetilde{f}^T(\theta)$ yields the gradient

$$\nabla J(\theta) = \mathbb{E}_{T \sim \mathcal{P}(\mathcal{T})} \nabla \widetilde{f}^T(\theta + \alpha \nabla \widetilde{f}^T(\theta))(\mathbf{I} + \alpha \nabla^2 \widetilde{f}^T(\theta)). \tag{9}$$

This is analogous to formulas obtained in e.g (Liu et al., 2019) for the policy gradient MAML. We can then approximate this gradient as an input to stochastic first-order methods. An example with standard SGD is shown in Algorithm 5.

**Data:** initial policy $\theta_0$, adaptation step size $\alpha$, meta step size $\beta$, number of queries $K$

1 **for** $t = 0, 1, \dots$ **do**
2    Sample $n$ tasks $T_1, \dots, T_n$;
3    **foreach** $T_i$ **do**
4      $\mathbf{d}_1^{(i)} \leftarrow \text{ESGRAD}(f^{T_i}, \theta_t, K, \sigma)$;
5      $\mathbf{H}^{(i)} \leftarrow \text{ESHESS}(f^{T_i}, \theta_t, K, \sigma)$;
6      $\theta_t^{(i)} \leftarrow \theta_t + \alpha \cdot \mathbf{d}_i$;
7      $\mathbf{d}_2^{(i)} \leftarrow \text{ESGRAD}(f^{T_i}, \theta_t^{(i)}, K, \sigma)$;
8    **end**
9    $\theta_{t+1} \leftarrow \theta_t + \frac{\beta}{n} \sum_{i=1}^n (\mathbf{I} + \alpha \mathbf{H}^{(i)}) \mathbf{d}_2^{(i)}$;
10 **end**

**Algorithm 5:** First Order ES-MAML

1 **ESHess** $(f, \theta, n, \sigma)$
   **inputs:** function $f$, policy $\theta$, number of perturbations $n$, precision $\sigma$
2    Sample i.i.d $\mathcal{N}(0, \mathbf{I})$ vectors $\mathbf{g}_1, \dots, \mathbf{g}_n$;
3    $v \leftarrow \frac{1}{n} \sum_{i=1}^n f(\theta + \sigma \mathbf{g}_i)$;
4    $\mathbf{H}^0 \leftarrow \frac{1}{n} \sum_{i=1}^n f(\theta + \sigma \mathbf{g}_i) \mathbf{g}_i \mathbf{g}_i^T$;
5    **return** $\frac{1}{\sigma^2}(\mathbf{H}^0 - v \cdot \mathbf{I})$;

**Algorithm 4:** Monte Carlo ES Hessian

A central problem, as discussed in (Rothfuss et al., 2019; Liu et al., 2019) is the estimation of $\nabla^2 \widetilde{f}^T(\theta)$. However, a simple expression exists for this object in the ES setting; it can be shown that

$$\nabla^2 \widetilde{f}^T(\theta) = \frac{1}{\sigma^2}(\mathbb{E}_{\mathbf{h} \sim \mathcal{N}(0, \mathbf{I})}[f^T(\theta + \sigma \mathbf{h}) \mathbf{h} \mathbf{h}^T] - \widetilde{f}^T(\theta) \mathbf{I}). \tag{10}$$

Note that for the vector $\mathbf{h}$, $\mathbf{h}^T$ is the transpose (and unrelated to tasks $T$). A basic MC estimator is shown in Algorithm 4. Given an independent estimator for $\nabla \widetilde{f}^{\mathcal{T}}(\theta + \alpha \nabla \widetilde{f}^T(\theta))$, we can then take the product to obtain an estimator for $\nabla J$.

### A.1.2 EXPERIMENTS WITH FIRST-ORDER ES-MAML

Unlike zero-order ES-MAML (Algorithm 3), the first-order ES-MAML explicitly builds an approximation of the Hessian of $f^T$. Given the literature on PG-MAML, we expect that estimating the Hessian $\nabla^2 \widetilde{f}^T(\theta)$ with Algorithm 4 without any control variates may have high variance. We compare two variants of first-order ES-MAML:

1. The full version (FO-Hessian) specified in Algorithm 5.

2. The 'first-order approximation' (FO-NoHessian) which ignores the term $\mathbf{I} + \alpha \nabla^2 \widetilde{f}^T(\theta)$ and approximates the MAML gradient as $\mathbb{E}_{T \sim \mathcal{P}(\mathcal{T})} \nabla \widetilde{f}^T(\theta + \alpha \nabla \widetilde{f}^T(\theta))$. This is equivalent to setting $\mathbf{H}^{(i)} = 0$ in line 5 of Algorithm 5.

The results on the four corner exploration problem (Section 4.1) and the Forward-Backward Ant, using Linear policies, are shown in Figure A1. On Forward-Backward Ant, FO-NoHessian actually outperformed FO-Hessian, so the inclusion of the Hessian term actually slowed convergence. On the four corners task, both FO-Hessian and FO-NoHessian have large error bars, and FO-Hessian slightly outperforms FO-NoHessian.

There is conflicting evidence as to whether the same phenomenon occurs with PG-MAML; (Finn et al., 2017, §5.2) found that on *supervised learning* MAML, omitting Hessian terms is competitive

Figure A1: Comparisons between the FO-Hessian and FO-NoHessian variants of Algorithm 5.

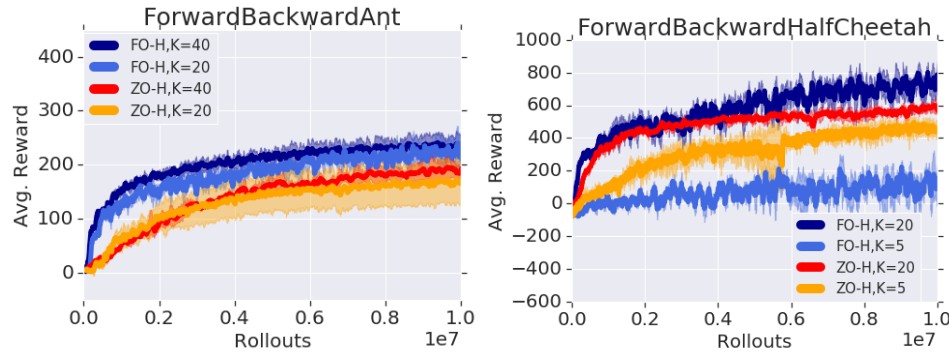

but slightly worse than the full PG-MAML, and does not report comparisons with and without the Hessian on RL MAML. (Rothfuss et al., 2019; Liu et al., 2019) argue for the importance of the second-order terms in proper credit assignment, but use heavily modified estimators (LVC, control variates; see Section 2) in their experiments, so the performance is not directly comparable to the 'naive' estimator in Algorithm 4. Our interpretation is that Algorithm 4 has high variance, making the Hessian estimates inaccurate, which can slow training on relatively 'easier' tasks like Forward-Backward walking but possibly increase the exploration on four corners.

We also compare FO-NoHessian against Algorithm 3 on Forward-Backward HalfCheetah and Ant in Figure A2. In this experiment, the two methods ran on servers with different number of workers available, so we measure the score by the total number of rollouts. We found that FO-NoHessian was slightly faster than Algorithm 3 when measured by rollouts on Ant, but FO-NoHessian had notably poor performance when the number of queries was low ($K = 5$) on HalfCheetah, and failed to reach similar scores as the others even after running for many more rollouts.

Figure A2: Comparisons between FO-NoHessian and Algorithm 3, by rollouts

## A.2 HANDLING ESTIMATOR BIAS

Since the adapted policy $U(\theta, T)$ generally cannot be evaluated exactly, we cannot easily obtain unbiased estimates of $f^T(U(\theta, T))$. This problem arises for both PG-MAML and ES-MAML.

We consider PG-MAML first as an example. In PG-MAML, the adaptation operator is $U(\theta, T) = \theta + \alpha \nabla_\theta \mathbb{E}_{\tau \sim \mathcal{P}_\mathcal{T}(\tau|\theta)}[R(\tau)]$. In general, we can only obtain an estimate of $\nabla_\theta \mathbb{E}_{\tau \sim \mathcal{P}_\mathcal{T}(\tau|\theta)}[R(\tau)]$ and not its exact value. However, the MAML gradient is given by

$$\nabla_\theta J(\theta) = \mathbb{E}_{\mathcal{T} \sim \mathcal{P}(\mathcal{T})}[\mathbb{E}_{\tau' \sim \mathcal{P}_\mathcal{T}(\tau'|\theta')}[\nabla_{\theta'} \log \mathcal{P}_\mathcal{T}(\tau'|\theta') R(\tau') \nabla_\theta U(\theta, T)]] \tag{11}$$

which requires exact sampling from the adapted trajectories $\tau' \sim \mathcal{P}_\mathcal{T}(\tau'|U(\theta, T))$. Since this is a nonlinear function of $U(\theta, T)$, we cannot obtain unbiased estimates of $\nabla J(\theta)$ by sampling $\tau'$ generated by an *estimate* of $U(\theta, T)$.

In the case of ES-MAML, the adaptation operator is $U(\theta, T) = \theta + \alpha \nabla \widetilde{f}(\theta, T) = \mathbb{E}_{\mathbf{h}} u(\theta, T; \mathbf{h})$ for $\mathbf{h} \sim \mathcal{N}(0, I)$, where $u(\theta, T; \mathbf{h}) = \theta + \frac{\alpha}{\sigma} f^{\mathcal{T}}(\theta + \sigma \mathbf{h}) \mathbf{h}$. Clearly, $f^T(u(\theta, T; \mathbf{h}))$ is not an unbiased estimator of $f^{\mathcal{T}}(U(\theta, T))$.

We may question whether using an unbiased estimator of $f^T(U(\theta, T))$ is likely to improve performance. One natural strategy is to reformulate the objective function so as to make the desired estimator unbiased. This happens to be the case for the algorithm E-MAML (Al-Shedivat et al., 2018), which treats the adaptation operator as an explicit function of $K$ sampled trajectories and "moves the expectation outside". That is, we now have an adaptation operator $U(\theta, T; \tau_1, \ldots, \tau_K)$, and the objective function becomes

$$\mathbb{E}_T[\mathbb{E}_{\tau_1, \ldots, \tau_k \sim \mathcal{P}_{\mathcal{T}}(\tau|\theta)} f^T(U(\theta, T; \tau_1, \ldots, \tau_K))] \tag{12}$$

An unbiased estimator for the E-MAML gradient can be obtained by sampling only from $\tau \sim \mathcal{P}_{\mathcal{T}}(\tau|\theta)$ (Al-Shedivat et al., 2018). However, it has been argued that by doing so, E-MAML does not properly assign credit to the pre-adaptation policy (Rothfuss et al., 2019). Thus, this particular mathematical strategy seems to be disadvantageous for RL.

The problem of finding estimators for function-of-expectations $f(\mathbb{E}X)$ is difficult and while general unbiased estimation methods exist (Blanchet et al., 2017), they are often complicated and suffer from high variance. In the context of MAML, ProMP compares the *low variance curvature* (LVC) estimator (Rothfuss et al., 2019), which is biased, against the unbiased DiCE estimator (Foerster et al., 2018), for the Hessian term in the MAML gradient, and found that the lower variance of LVC produced better performance than DiCE. Alternatively, control variates can be used to reduce the variance of the DiCE estimator, which is the approach followed in (Liu et al., 2019).

In the ES framework, the problem can also be formulated to avoid exactly evaluating $U(\cdot, T)$, and hence circumvents the question of estimator bias. We observe an interesting connection between MAML and the *stochastic composition* problem. Let us define $u_{\mathbf{h}}(\theta, T) = u(\theta, T; \mathbf{h})$ and $f_{\mathbf{g}}^T(\theta) = f^T(\theta + \sigma \mathbf{g})$. For a given task $T$, the MAML reward is given by

$$\widetilde{f}^{\mathcal{T}}(U(\theta, T)) = \widetilde{f}^T[\mathbb{E}_{\mathbf{h}} u_{\mathbf{h}}(\theta, T)] = \mathbb{E}_{\mathbf{g}} f_{\mathbf{g}}^T(\mathbb{E}_{\mathbf{h}} u_{\mathbf{h}}(\theta, T)). \tag{13}$$

This is a two-layer nested stochastic composition problem with outer function $\widetilde{f}^T = \mathbb{E}_{\mathbf{g}} f_{\mathbf{g}}^T$ and inner function $U(\cdot, T) = \mathbb{E}_{\mathbf{h}} u_{\mathbf{h}}(\cdot, T)$. An accelerated algorithm (ASC-PG) was developed in (Wang et al., 2017)] for this class of problems. While neither $f_{\mathbf{g}}^T$ nor $u_{\mathbf{h}}(\cdot, T)$ is smooth, which is assumed in (Wang et al., 2017), we can verify that the crucial content of the assumptions hold:

1. $\mathbb{E}_{\mathbf{h}} u_{\mathbf{h}}(\theta, T) = U(\theta, T)$
2. We can define two functions

$$\zeta_{\mathbf{g}}^T(\theta) = \frac{1}{\sigma} f_{\mathbf{g}}^T(\theta) \mathbf{g}, \quad \xi_{\mathbf{h}}^T(\theta) = \mathbf{I} + \frac{\alpha}{\sigma^2} (f_{\mathbf{h}}^T(\theta) \mathbf{h} \mathbf{h}^T - f_{\mathbf{h}}^T(\theta) \mathbf{I})$$

such that for any $\theta_1, \theta_2$,

$$\mathbb{E}_{\mathbf{g}, \mathbf{h}}[\xi_{\mathbf{h}}^T(\theta_1) \zeta_{\mathbf{g}}^T(\theta_2)] = JU(\theta_1, T) \nabla \widetilde{f}^T(\theta_2)$$

where $JU$ denotes the Jacobian of $U(\cdot, T)$, and $\mathbf{g}, \mathbf{h}$ are independent vectors sampled from $\mathcal{N}(0, \mathbf{I})$. This follows immediately from equation 4 and equation 10.

The ASC-PG algorithm does not immediately extend to the full MAML problem, as upon taking an outer expectation over $T$, the MAML reward $J(\theta) = \mathbb{E}_T \mathbb{E}_{\mathbf{g}} f_{\mathbf{g}}^T(\mathbb{E}_{\mathbf{h}} u_{\mathbf{h}}(\theta, T))$ is no longer a stochastic composition of the required form. In particular, there are conceptual difficulties when the number of tasks in $\mathcal{T}$ is infinite. However, it can be used to solve the MAML problem for each task within a consensus framework, such as consensus ADMM (Hong et al., 2016).

## A.3 EXTENSIONS OF ES

In this section, we discuss several general techniques for improving the basic ES gradient estimator (Algorithm 1). These can be applied both to the ES gradient of the meta-training (the 'outer loop' of Algorithm 3), and more interestingly, to the adaptation operator itself. That is, given $U(\theta, T) =$

$\theta + \alpha \nabla \widetilde{f}_\sigma^T(\theta)$, we replace the estimation of $U$ by ESGRAD on line 4 of Algorithm 3 with an improved estimator of $\nabla \widetilde{f}_\sigma^T(\theta)$, which even may depend on data collected during the meta-training stage. Many techniques exist for reducing the variance of the estimator such as Quasi Monte Carlo sampling (Choromanski et al., 2018). Aside from variance reduction, there are also methods with special properties.

### A.3.1 ACTIVE SUBSPACES

Active Subspaces is a method for finding a low-dimensional subspace where the contribution of the gradient is maximized. Conceptually, the goal is to find and update on-the-fly a low-rank subspace $\mathcal{L}$ so that the projection $\nabla f^T(\theta)_{\mathcal{L}}$ of $\nabla f^T(\theta)$ into $\mathcal{L}$ is maximized and apply $\nabla f^T(\theta)_{\mathcal{L}}$ instead of $\nabla f^T(\theta)$. This should be done in such a way that $\nabla f^T(\theta)$ does not need to be computed explicitly. Optimizing in lower-dimensional subspaces might be computationally more efficient and can be thought of as an example of guided ES methods, where the algorithm is guided how to explore space in the anisotropic way, leveraging its knowledge about function optimization landscape that it gained in the previous steps of optimization. In the context of RL, the active subspace method ASEBO (Choromanski et al., 2019b) was successfully applied to speed up policy training algorithms. This strategy can be made data-dependent also in the MAML context, by learning an optimal subspace using data from the meta-training stage, and sampling from that subspace in the adaptation step.

### A.3.2 REGRESSION-BASED OPTIMIZATION

Regression-Based Optimization (RBO) is an alternative method of gradient estimation. From Taylor series expansion we have $f(\theta + \mathbf{d}) - f(\theta) = \nabla f(\theta)^T \mathbf{d} + O(\|\mathbf{d}\|^2)$. By taking multiple finite difference expressions $f(\theta + \mathbf{d}) - f(\theta)$ for different $\mathbf{d}$, we can recover the gradient by solving a regularized regression problem. The regularization has an additional advantage - it was shown that the gradient can be recovered even if a substantial fraction of the rewards $f(\theta + \mathbf{d})$ are corrupted (Choromanski et al., 2019c). Strictly speaking, this is not based on the Gaussian smoothing as in ES, but is another method for estimating gradients using only zero-th order evaluations.

### A.3.3 EXPERIMENTS

We present a preliminary experiment with RBO and ASEBO gradient adaptation in Figure A3. To be precise, the algorithms used are identical to Algorithm 3 except that in line 4, $\mathbf{d}^{(i)} \leftarrow$ ESGRAD is replaced by $\mathbf{d}^{(i)} \leftarrow$ RBO (yielding RBO-MAML) and $\mathbf{d}^{(i)} \leftarrow$ ASEBO (yielding ASEBO-MAML) respectively.

Figure A3: RBO-MAML and ASEBO-MAML compared to ES-MAML.

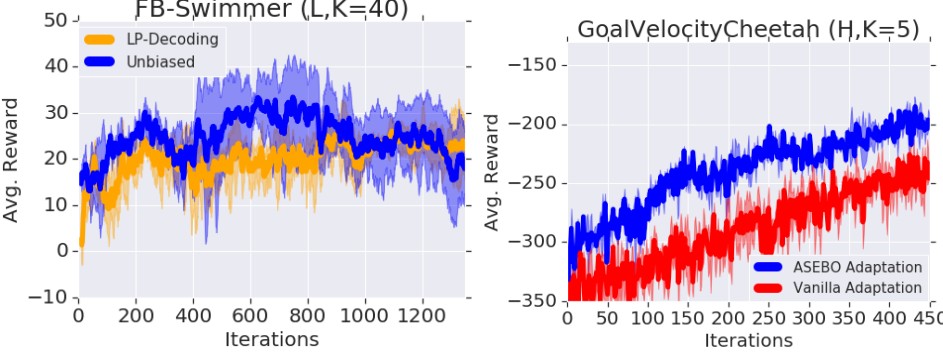

On the left plot, we test for noise robustness on the Forward-Backward Swimmer MAML task, comparing standard ES-MAML (Algorithm 3) to RBO-MAML. To simulate noisy data, we randomly corrupt 25% of the queries $f^T(\theta + \sigma g)$ used to estimate the adaptation operator $U(\theta, T)$ with an enormous additive noise. This is the same type of corruption used in (Choromanski et al., 2019c).

Interestingly, RBO does not appear to be more robust against noise than the standard MC estimator, which suggests that the original ES-MAML has some inherent robustness to noise.

On the right plot, we compare ASEBO-MAML to ES-MAML on the Goal-Velocity HalfCheetah task in the low-$K$ setting. We found that when measured in iterations, ASEBO-MAML outperforms ES-MAML. However, ASEBO requires additional linear algebra operations and thus uses significantly more wall-clock time (not shown in plot) per iteration, so if measured by real time, then ES-MAML was more effective.

## A.4  NAVIGATION-2D EXPLORATION TASK

*Navigation-2D* (Finn et al., 2017) is a classic environment where the agent must explore to adapt to the task. The agent is represented by a point on a 2D square, and at each time step, receives reward equal to its distance from a given target point on the square. Note that unlike the four corners and six circles tasks, the reward for Navigation-2D is dense. We visualize the differing exploration strategies learned by PG-MAML and ES-MAML in Figure A4. Notice that PG-MAML makes many tiny movements in multiple directions to 'triangulate' the target location using the differences in reward for different state-action pairs. On the other hand, ES-MAML learns a meta-policy such that each perturbation of the meta-policy causes the agent to move in a different direction (represented by red paths), so it can determine the target location from the total rewards of each path.

Figure A4: Comparing the exploration behavior of PG-MAML and ES-MAML on the Navigation-2D task. We use $K = 20$ queries for each algorithm.

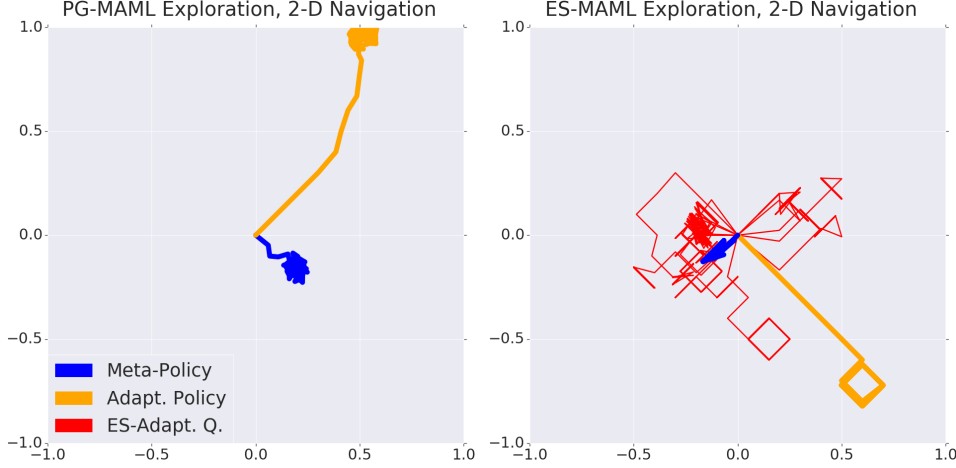

## A.5   PG-MAML RL BENCHMARKS

In Figure A5, we compare ES-MAML and PG-MAML on the Forward-Backward and Goal-Velocity tasks for HalfCheetah, Swimmer, Walker2d, and Ant, using the same values of $K$ that were used in the original experiments of (Finn et al., 2017).

Figure A5: Comparisons between ES-MAML and PG-MAML using the queries $K$ from (Finn et al., 2017).

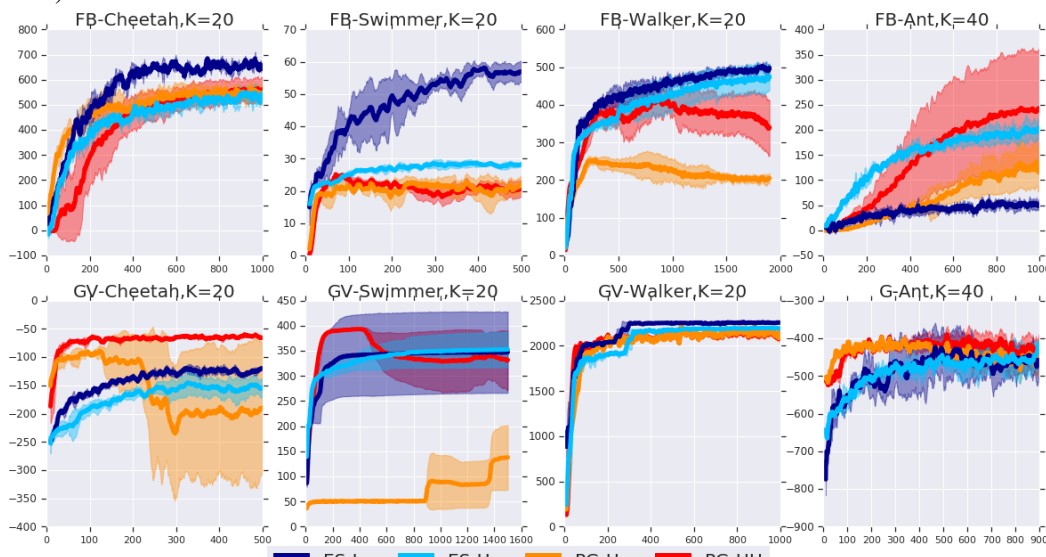

## A.6   REGRESSION AND SUPERVISED LEARNING

MAML has also been applied to supervised learning. We demonstrate ES-MAML on *sine regression* (Finn et al., 2017), where the task is to fit a sine curve $f$ with unknown amplitude and phase given a set of $K$ pairs $(x_i, f(x_i))$. The meta-policy must be able to learn that all of tasks have a common periodic nature, so that it can correctly adapt to an unknown sine curve outside of the points $x_i$.

For regression, the loss is the mean-squared error (MSE) between the adapted policy $\pi_\theta(x)$ and the true curve $f(x)$. Given data samples $\{(x_i, f(x_i)\}_{i=1}^K$, the empirical loss is $L(\theta) = \frac{1}{K} \sum_{i=1}^K (f(x_i) - \pi_\theta(x_i))^2$. Note that unlike in reinforcement learning, we *can* exactly compute $\nabla L(\theta)$; for deep networks, this is by automatic differentiation. Thus, we opt to use Tensorflow to compute the adaptation operator $U(\theta, T)$ in Algorithm 3. This is in accordance with the general principle that when gradients are available, it is more efficient to use the gradient than to approximate it by a zero-order method (Nesterov & Spokoiny, 2017).

We show several results in Figure A6. The adaptation step size is $\alpha = 0.01$, which is the same as in (Finn et al., 2017). For comparison, (Finn et al., 2017) reports that PG-MAML can obtain a loss of $\approx 0.5$ after one adaptation step with $K = 5$, though it is not specified how many iterations the meta-policy was trained for. ES-MAML approaches the same level of performance, though the number of training iterations required is higher than for the RL tasks, and surprisingly high for what appears to be a simpler problem. This is likely again a reflection of the fact that for problems such as regression where the gradients are available, it is more efficient to use gradients.

As an aside, this leads to a related question of the correct interpretation of the query number $K$ in the supervised setting. There is a distinction between obtaining a data sample $(x_i, f(x_i))$, and doing a computation (such as a gradient) using that sample. If the main bottleneck is collecting the data $\{(x_i, f(x_i)\}$, then we may be satisfied with any algorithm that performs any number of operations on the data, as long as it uses only $K$ samples. On the other hand, in the (on-policy) RL setting, samples cannot typically be 're-used' to the same extent, because rollouts $\tau$ sampled with a given

Figure A6: The MSE of the adapted policy, for varying number of gradient steps and query number $K$. Runs are averaged across 3 seeds.

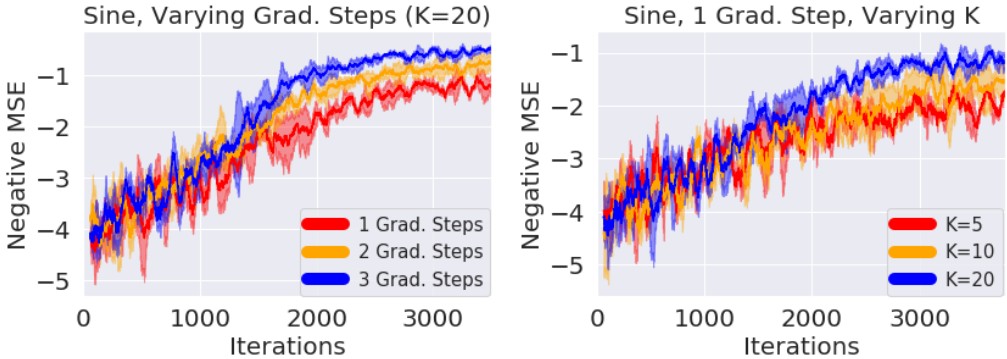

policy $\pi_\theta$ follow an unknown distribution $\mathcal{P}(\tau|\theta)$ which reduces their usefulness away from $\theta$. Thus, the corresponding notion to rollouts in the SL setting would be the number of backpropagations (for PG-MAML) or perturbations (for ES-MAML), but clearly these have different relative costs than doing simulations in RL.

## A.7 Hyperparameters and Setups

### A.7.1 Environments

Unless otherwise explicitly stated, we default to $K = 20$ and horizon = 200 for all RL experiments. We also use the standard reward normalization in (Mania et al., 2018), and use a global state normalization (i.e. the same mean, standard deviation normalization values for MDP states are shared across workers).

For the Ant environments (Goal-Position Ant, Forward-Backward Ant), there are significant differences in weighting on the auxiliary rewards such as control costs, contact costs, and survival rewards across different previous work (e.g. those costs are downweighted in (Finn et al., 2017) whereas the coefficients are vanilla Gym weightings in (Liu et al., 2019)). These auxiliary rewards can lead to local minima, such as the agent staying stationary to collect the survival bonus which may be confused with movement progress when presenting a training curve. To make sure the agent is explicitly performing the required task, we opted to remove such costs in our work and only present the main goal-distance cost and forward-movement reward respectively.

For the other environments, we used default weightings and rewards, since they do not change across previous works.

### A.7.2 ES-MAML Hyperparameters

Let $N$ be the number of possible distinct tasks possible. We sample tasks without replacement, which is important if $N \ll 5$, as each worker performs adaptations on all possible tasks.

For standard ES-MAML (Algorithm 3), we used the following settings.

| Setting | Value |
|---|---|
| (Total Workers, # Perturbations, # Current Evals) | (300, 150, 150) |
| (Train Set Size, Task Batch Size, Test Set Size) | (50,5,5) or (N,N,N) |
| Number of rollouts per parameter | 1 |
| Number of Perturbations per worker | 1 |
| Outer-Loop Precision Parameter | 0.1 |
| Adaptation Precision Parameter | 0.1 |
| Outer-Loop Step Size | 0.01 |
| Adaptation Step Size ($\alpha$) | 0.05 |
| Hidden Layer Width | 32 |
| ES Estimation Type | Forward-FD |
| Reward Normalization | True |
| State Normalization | True |

For ES-MAML and PG-MAML, we took 3 seeded runs, using the default TRPO hyperparameters found in (Liu et al., 2019).

