# OpenReview forum: "ES-MAML: Simple Hessian-Free Meta Learning"
_ICLR.cc/2020/Conference — Accept (Poster)_

### Official Review · AnonReviewer3 · 2019-10-08
**Official Blind Review #3**

**Rating:** 1

**Review:**

The paper proposes ES for the task of Model agnostic meta learning. Instead of the gradient-approximation which requires computing a hessian matrix, MC samples from a search distribution are used to estimate a search direction. The approach is validated on a number of experiments.

Unfortunatly, I am unable to accept this paper for a number of reasons. Mainly that the ES used is inferior and the constant step-size used can have a major effect on the experimental outcome.

Almost all proper ES literature with real working ES algorithms are missing and ESGrad is more than 20 years behind SOTA in the field. Since ES are central to the paper, an algorithm that would not even be considered a baseline at any conference in that field is difficult to accept.
The reason for this is that nowadays all ES use dynamic sample-variances based on progress measures, e.g. Cumulative step-size adaptation and Two-Point-Adaptation as the SOTA. Without this, it can be very difficult to find reasonable solutions.

Most important missing references from the ES-field in this context:

1. and most importantly The original ES-based RL paper:
Heidrich-Meisner, Verena, and Christian Igel. "Neuroevolution strategies for episodic reinforcement learning." Journal of Algorithms 64.4 (2009): 152-168.

2. CMA-ES and NES
Hansen, N., Müller, S. D., & Koumoutsakos, P. (2003). Reducing the time complexity of the derandomized evolution strategy with covariance matrix adaptation (CMA-ES). Evolutionary computation, 11(1), 1-18.
Krause, O., Arbonès, D. R., & Igel, C. (2016). CMA-ES with optimal covariance update and storage complexity. In Advances in Neural Information Processing Systems (pp. 370-378).
Wierstra, D., Schaul, T., Peters, J., & Schmidhuber, J. (2008, June). Natural evolution strategies. In 2008 IEEE Congress on Evolutionary Computation (IEEE World Congress on Computational Intelligence) (pp. 3381-3387). IEEE.

3. Review of SOTA in large-scale ES:
Varelas, K., Auger, A., Brockhoff, D., Hansen, N., ElHara, O. A., Semet, Y., ... & Barbaresco, F. (2018, September). A comparative study of large-scale variants of CMA-ES. In International Conference on Parallel Problem Solving from Nature (pp. 3-15). Springer, Cham.

4. Recent developments for noisy functions (also references other relevant algorithms with noise-handling)
Krause, O. (2019, July). Large-scale noise-resilient evolution-strategies. In Proceedings of the Genetic and Evolutionary Computation Conference (pp. 682-690). ACM.

Section 3.2
-Why should in (7) the same sigma be used as in (6)? Sigma, alpha etc should be learnable parameters learned by the outer ES.
-3.3.2: you are writing below (1) that rollouts come from a distribution, i.e. are stochastic. How would you implement a hill-climber in the stochastic setting? e.g. consider the case when the rewards are heavy-tailed.
- using a hill-climber goes completely against the SOTA in ES which showed repeatedly over the last 20 years that hill-climbing is inferior, especially in larger dimension search-spaces (>100).

Experiments:
- I am not an expert of MAML, but i would not consider this as different tasks, just as different environments for the same task. i.e. a circular running strategy should be optimal for all environments. but when considering different tasks, we would consider different policies to be optimal.
- The experiments use the same hyper parameters for all variants. However, i am not sure this is a fair comparison. E.g. HC has way more spread over the search-space than the other two methods for a given sigma, with following sample steps allowing for fixing the "too large" or "too small" spread.
Since the graph of the objective function is flat in a large area of the search space, the additional exploration through stocasticity alone might explain the results of Figure 1. In this case, the result would be pretty artificial, because real ES would adapt their step-size.
- Similar holds for the number of samples used by the outer ES (n, but named differently in th appendix?). The gradient-based approaches might require a lot more initial points with a smaller K , especially on the flat surfaces of the objectives.
- In Figure 3, middle image, why does the green curve appear to have decreasing performance after iteration 200?
- Figure 3/ 4.2 why do the three settings have different values for number of iterations and K? Why does L-DPP only appear in the third task?
-Section 4.3 and Figure 4: why is there no L-PG and HH-ES? the only curve which is is available for both algorithms has the same performance.

**Experience Assessment:**

I have published in this field for several years.

**Review Assessment: Checking Correctness Of Derivations And Theory:**

I assessed the sensibility of the derivations and theory.

**Review Assessment: Checking Correctness Of Experiments:**

I assessed the sensibility of the experiments.

**Review Assessment: Thoroughness In Paper Reading:**

I read the paper at least twice and used my best judgement in assessing the paper.

---

> ### Author Response · Authors · 2019-11-07
> **Author Response to Official Blind Review #3, part 1**
>
> Our response is long, so we have included a summary and then a detailed response.
>
> Important point on terminology:
> We believe the review is inappropriately conflating the "ES" algorithms in our paper with other methods called "evolution", particularly CMA-ES. While these are related, neither is 'extended' from the other, so e.g. it is not correct to imply that any modifications developed to speed up CMA-ES can also be applied to our ES.
> To clearly distinguish these, we will refer to our algorithm as "ES/ARS" (since it has been called Random Search in some papers), and since the review mainly discusses CMA-ES, we will use "CMA-ES" as an umbrella term for their referenced methods.
>
> SUMMARY:
> The main criticism seems to be that the ES/ARS algorithm used in our paper is not SOTA. We do not believe this to be valid for these reasons:
> 1. There are recent papers in RL using ES/ARS which achieve SOTA results [2,3,5,6], so we strongly disagree with the blanket statement that "the ES used is inferior".
> 2. Moreover, the review does not address MAML or meta-learning, the specific problem class we are solving.
> 3. Though ES/ARS and CMA-ES have similarities, they are different methods and correspond to different formulations of the loss function. Techniques such as dynamic sampling variance for CMA-ES are not readily justified in the theoretical framework for ES/ARS. So it is not the case that our ES/ARS is somehow an 'obsolete' version of CMA-ES.
> 4. Our experiments show that a simple ES/ARS algorithm (with constant step-size) is *already* competitive with the existing policy gradient methods on MAML. Thus, if it were true that ES/ARS can be improved even more by using a step-size schedule and other heuristics, then this should evidence *in favor* of ES/ARS for meta-learning, not a point against it. That is to say, we are arguing that our ES-MAML has advantages over the current PG-MAML, not that we have designed an "optimal" ES.
>
>
> DETAILED COMMENTS:
>
> >> "Almost all proper ES literature with real working ES algorithms are missing and ESGrad is more than 20 years behind SOTA in the field. Since ES are central to the paper, an algorithm that would not even be considered a baseline at any conference in that field is difficult to accept."
>
> The claim that "ES Grad is more than 20 years behind SOTA" and "would not even be considered a baseline at any conference" is clearly contradicted by the recently published papers [2,3,5,6].
> We object to the assertion that ES/ARS is not "real working ES"  and to the reviewer's complete dismissal of related work as not being "proper ES literature".
>
> >> "The reason for this is that nowadays all ES use dynamic sample-variances based on progress measures, e.g. Cumulative step-size adaptation and Two-Point-Adaptation as the SOTA. Without this, it can be very difficult to find reasonable solutions."
>
> Though there are similarities, ES/ARS is not the same as CMA-ES. ES/ARS algorithms are based on a particular Gaussian smoothing of the loss function, which rigorously establishes that the estimator does yield a stochastic gradient of the smoothed loss. Other elements such as dynamic sample variance can (for specific cases) be viewed as covariance adaptation or as arising from the natural gradient [10], but are not theoretically justified given the objective for ES/ARS. Moreover, many techniques such as the two-point-adaptation are heuristics and currently lack a theoretical basis, and it is not our goal to use every heuristic in our algorithm.
>
> As to whether such elements are necessary for good performance, [7,8] (using population search) report SOTA results on RL using a simple evolutionary algorithm which does not employ covariance adaptation. So it is not a given that such elements are essential for good performance.
>
> We also wish to point out again that our experiments show that a simple version of ES/ARS is already competitive with policy gradients on MAML. If additional heuristics can make ES/ARS even better, then that further supports the importance of investigating ES/ARS for meta-learning. Our experiments indicate that we can already obtain reasonable solutions using ES/ARS without those modifications.
>
> Regarding "ES Gradient", we want to clarify that the experiments are not using equation (4), which is the algorithm in Box 1, to estimate the ES-gradient; this equation is provided to explain the theory of the smoothing. As we discuss in the paragraph immediately following, in practice the Forward FD or antithetic version of the estimator is used. The antithetic formula has some resemblance to "Two-Point-Adaptation" (though, note in the ES/ARS context, it is a method for reducing the estimator variance, not for modifying the step-size). Our experimental results use the Forward FD estimator (as described in the hyperparameters table), not eq (4).

---

> > ### Author Response · Authors · 2019-11-07
> > **Author Response to Official Blind Review #3, part 2**
> >
> > >> "[references]"
> >
> > Thanks for the references. We have revisited them and added several citations, in particular those of NES, and we have added some discussion of CMA-ES as an alternative to hill climbing as an adaptation operator. However, we would still hold that works focusing *purely* on CMA-ES are not central to ES/ARS for reinforcement learning to the same degree that e.g [2,4] are.
> >
> > >> "Why should in (7) the same sigma be used as in (6)? Sigma, alpha etc should be learnable parameters learned by the outer ES."
> >
> > It is not required that the inner ES-gradient \sigma must be the same as the outside \sigma. However, making it a learned parameter introduces the same conceptual issues as varying it over time (see above) and is not trivial.
> >
> > >> "3.3.2: you are writing below (1) that rollouts come from a distribution, i.e. are stochastic. How would you implement a hill-climber in the stochastic setting? e.g. consider the case when the rewards are heavy-tailed."
> >
> > For stochastic rewards, some of the queries can be used to obtain better estimates of the reward function at the same point. Yes, this can be difficult to estimate accurately when the distribution has heavy tails. Empirically, we found that hill climbing was effective on the MAML benchmark problems.
> >
> > Our motivation for experimenting with hill climbing was not to argue that hill climbing is always a great adaptation operator. Rather, we meant to demonstrate that ES-MAML can use non-smooth adaptation operators in a theoretically principled way (in this case, the local search operator), which is an advantage it has over policy gradient MAML.
> >
> > >> "using a hill-climber goes completely against the SOTA in ES which showed repeatedly over the last 20 years that hill-climbing is inferior, especially in larger dimension search-spaces (>100)."
> >
> > Note that we are using hill climbing as an inner adaptation operator, not for optimizing the outer ES loop. Our motivation is not to argue that hill climbing is always a great adaptation operator, rather it is an example of a nonsmooth adaptation operator. Since ES-MAML can handle non-smooth operators, we could also use more refined versions of CMA-ES in the inner operator, but this is tangential to the main goals of the paper.
> >
> > >> "The experiments use the same hyper parameters for all variants. However, i am not sure this is a fair comparison. E.g. HC has way more spread over the search-space than the other two methods for a given sigma, with following sample steps allowing for fixing the "too large" or "too small" spread.
> > Since the graph of the objective function is flat in a large area of the search space, the additional exploration through stochasticity alone might explain the results of Figure 1. In this case, the result would be pretty artificial, because real ES would adapt their step-size."
> >
> > The primary comparison we wish to highlight is between ES-MAML and PG-MAML, not between variants of ES-MAML. There are indeed techniques for improving the performance of ES-MAML with ES-gradient adaptation (Algorithm 3), but whether we can make Algorithm 3 better than Algorithm 2 + HC is again tangential.
> >
> > Since in ES/ARS the estimator provides a stochastic gradient, Algorithm 3 is taking a SGD step with the fixed step-size \alpha. This is completely standard for MAML (the PG-MAML takes a policy gradient step of fixed step-size). Furthermore, we found that this was the optimal \alpha parameter - higher or lower \alpha gave worse performance.
> > The question of whether this is "real ES" is only meaningful if one is interpreting ES/ARS as a type of CMA-ES, but this is derived from different principles.
> >
> > >> "- In Figure 3, middle image, why does the green curve appear to have decreasing performance after iteration 200?
> > - Figure 3/ 4.2 why do the three settings have different values for number of iterations and K? Why does L-DPP only appear in the third task?
> > -Section 4.3 and Figure 4: why is there no L-PG and HH-ES? the only curve which is is available for both algorithms has the same performance."
> >
> > In Fig 3, the performance of L-HC has plateaued at a near-optimality. There are fluctuations in the reward because the algorithm is stochastic, but we did not observe it to be systematically decreasing.
> >
> > In Fig 3 and 4, the settings are different because the environments and tasks are different. This is also standard for MAML (e.g., one typically uses a larger K for the Ant agent which is more difficult to learn [1]). The key is that within each environment, the algorithms are given the same K.
> >
> > In Figure 4, the reason for omitting L-PG is that PG-MAML does not train to a reasonable performance on linear policies. We found this in our experiments, and it has also been observed in previous MAML work [9]. Note that on the Forward-Backward Walker2d, ES-MAML with both architectures had better performance than PG-MAML; with the Swimmer agent, the hidden layer policies were indeed similar, but the linear policy was clearly above the others.

---

> > > ### Author Response · Authors · 2019-11-07
> > > **Author Response to Official Blind Review #3, references**
> > >
> > > References:
> > > [1] Chelsea Finn, Pieter Abbeel, and Sergey Levine. Model-agnostic meta-learning for fast adaptation of deep networks. ICML 2017
> > >
> > > [2] Tim Salimans, Jonathan Ho, Xi Chen, Szymon Sidor, and Ilya Sutskever. Evolution strategies as a scalable alternative to reinforcement learning. arXiv:1703.03864, 2017.
> > >
> > > [3] Horia Mania, Aurelia Guy, and Benjamin Recht. Simple random search provides a competitive approach to reinforcement learning. NIPS 2018
> > >
> > > [4] Yurii Nesterov and Vladimir Spokoiny. Random gradient-free minimization of convex functions. Foundations of Computational Mathematics, 17(2):527–566, 2017.
> > >
> > > [5] Krzysztof Choromanski, Mark Rowland, Vikas Sindhwani, Richard E. Turner, and Adrian Weller. Structured evolution with compact architectures for scalable policy optimization. ICML 2018
> > >
> > > [6] Krzysztof Choromanski, Aldo Pacchiano, Jack Parker-Holder, Yunhao Tang, Deepali Jain, Yuxiang Yang, Atil Iscen, Jasmine Hsu, and Vikas Sindhwani. Provably robust blackbox optimization for reinforcement learning. CoRL 2019
> > >
> > > [7] Felipe Petroski Such, Vashisht Madhavan, Edoardo Conti, Joel Lehman, Kenneth Stanley, and Jeff Clune. Deep neuroevolution: Genetic algorithms are a competitive alternative for training deep neural networks for reinforcement learning. arXiv:1712.06567, 2017.
> > >
> > > [8] Sebastian Risi and Kenneth Stanley. Deep neuroevolution of recurrent and discrete world models. arXiv:1906.08857, 2019.
> > >
> > > [9] Chelsea Finn, Sergey Levine. Meta-Learning and Universality: Deep Representations and Gradient Descent can Approximate any Learning Algorithm. arXiv:1710.11622, 2017.
> > >
> > > [10] Daan Wierstra, Tom Schaul, Jan Peters, and Juergen Schmidhuber. Natural Evolution Strategies. 2008 IEEE Congress on Evolutionary Computation

---

### Official Review · AnonReviewer1 · 2019-10-23
**Official Blind Review #1**

**Rating:** 6

**Review:**

This paper proposes a method, ES-MAML, for optimizing the Model Agnostic Meta Learning (MAML) objective by using Evolution Strategies (ES) gradients instead of policy gradients (PG) as in the previous approaches in the literature. As a result, the use of ES avoids the need of second-order derivative estimation resulted from PG in computing the gradients of the MAML objective; second-order derivatives in MAML are known to be tricky for proper estimation. They also explore ES-MAML with different advanced adaptation operators to improve the ES gradient estimator. They perform empirical study to demonstrate the benefits of ES-MAML as compared with PG-MAML. In particular, they evaluate the comparable algorithms (ES-MAML and variants vs PG-MAML) in terms of exploratory behaviors in sparse-reward environments, adaptation ability, the stability of deterministic policies in unstable environments, and low-K benchmarks. The experimental results are rigorous and promising. They also discuss several potential extensions to ES-MAML in the appendix.

Regarding the theoretical and algorithmic contributions, this paper combines existing techniques from ES and gradient estimators to make ES gradients work for MAML. Thus, I feel that the paper does not provide significantly new results on these dimensions. For example, the substitution of policy gradient for evolution strategies in Eq. (6) is straightforward, and there is no theoretical justification for the choice of algorithmic designs made in the paper.  However, given that the paper attempts to address an important problem (stably optimizing the MAML objective) with interesting perspective (using ES), that the proposed methods are well developed and extended, and that rigorous experiments to evaluate the proposed methods are provided, this paper could be an interesting contribution to the conference where it can encourage different perspective beyond the gradient policy view for MAML problems.

Questions and comments.

1. On page 3, with reference to the text “These issues: the difficulty of estimating the Hessian term (3), the typically high variance of ∇θJ(θ) for policy gradient algorithms in general, and the unsuitability of stochastic policies in some domains, lead us to the proposed method ES-MAML in Section 3.” I agree that the use of ES gradients avoids the need of second-order derivative estimation; however I am not very sure if we could say that ES-MAML here can address the high variance issue of PG given that ES can also suffer from high variance and that there is a rich literature in reducing variance of PG.

2. Could you clarify which version of PG-MAML was used as the baseline in your experiments? Is this the “vanilla” version from Eq. (2) without any variance reduction techniques (e.g., Rothfuss et al. (2019), Liu et al. (2019)) or did you include one of the variance reduction techniques to the baseline PG-MAML?

3. In section 4.2, with reference to the text “one of the main benefits of ES is due to its ability to train compact linear policies, which can outperform hidden-layer policies”, could you clarify what did this text mean? Did you mean that compact linear policies are better than hidden-layer policies for MAML, or does it mean that ES is not good at training hidden-layer policies, so it can train linear policies better than hidden-layer policies?

Minor comments.
1. Page 2, R has not been introduced.
2. Page 3, Section 3.1: does F mean f?

**Experience Assessment:**

I have read many papers in this area.

**Review Assessment: Checking Correctness Of Derivations And Theory:**

I assessed the sensibility of the derivations and theory.

**Review Assessment: Checking Correctness Of Experiments:**

I assessed the sensibility of the experiments.

**Review Assessment: Thoroughness In Paper Reading:**

I read the paper at least twice and used my best judgement in assessing the paper.

---

> ### Author Response · Authors · 2019-11-09
> **Author Response to Official Blind Review #1, Part 1**
>
> Thank you for your detailed review and feedback. We provide detailed answers to your questions below.
>
> >> "Thus, I feel that the paper does not provide significantly new results on these dimensions. For example, the substitution of policy gradient for evolution strategies in Eq. (6) is straightforward, and there is no theoretical justification for the choice of algorithmic designs made in the paper."
>
> Indeed, the formulation of MAML using ES via eq. (6) is very straightforward! We see this as an advantage, since it clearly shows that ES can be used to attack MAML, and the resulting algorithm is conceptually very simple but has nice theoretical properties (notably, eliminating the explicit derivative calculations). The ES-MAML algorithm then does not have many 'knobs' to adjust. Could you kindly clarify which parts of the algorithm design you feel have not been justified adequately?
>
> There are some other theoretical ideas in our paper that we believe are novel, and deserve greater attention within meta-learning. To highlight these:
>
> - Exploration in sparse environments
> A fundamental aspect of PG-MAML in meta-learning is that exploration is off-loaded to the meta-policy. The meta-policy must use its K trajectories to generate (s,a,s',r) samples for its replay buffer, and in the sparse-reward case, the hope is that some of these samples achieved nonzero reward. For a single policy, effective exploration requires it to have high entropy.
> In contrast, exploration in the parameter space with ES-MAML allows the meta-policy to use different 'styles' of exploration arising from different perturbations. This yields a richer set of behaviors and makes the meta-policy itself more stable. We discuss this further towards the end of the introduction, and in 4.1.
>
> - Estimation of meta-learning gradients
> There are a lot of challenges in correctly estimating the gradient of the composite MAML loss function. One which has been studied extensively is the problem of estimating the Hessian in PG-MAML (hence the DiCE estimator, LVC, T-MAML, etc.). Another which has received much less attention is that we still need to evaluate rewards under the *adapted trajectories*, i.e., under the distribution of trajectories after task adaptation. However, since we only have *estimates* of the policy after adaptation, the resulting estimates of any quantities based on the adapted trajectory are quite likely to be biased. This issue affects both PG-MAML and ES-MAML, and related to the general problem of unbiased estimation of functions-of-expectations, which is difficult. We discuss this topic in Appendix A.2.
>
> We also discuss the estimation of Hessians using ES in Appendix A.1. We found empirically that the zero-order ES-MAML is more effective than a 'first-order' ES-MAML using an ES-Hessian, so it is of marginal gain for the meta-learning problem under consideration. However our derivation of the ES Hessian estimator is, to the best of our knowledge, novel.
>
> >> "I agree that the use of ES gradients avoids the need of second-order derivative estimation; however I am not very sure if we could say that ES-MAML here can address the high variance issue of PG given that ES can also suffer from high variance and that there is a rich literature in reducing variance of PG. "
>
> Yes, it is true that there is a rich literature for reducing the variance of PG. In the case of PG-MAML, there is also extensive literature on reducing variance, specifically of the Hessian component. Our belief is that avoiding explicit dependence on the estimated Hessian is still important, because the concentration properties of stochastic Hessians are generally much weaker than stochastic gradients. So while one may use LVC or control variates, it is an important alternative to sidestep the Hessian estimation altogether.
>
> We also agree that the variance of ES methods can be problematic. There is also literature on reducing variance of ES gradients. The most simple method for instance is to use the antithetic estimator, but there are more sophisticated modifications such as orthogonal sampling and control variates [4,5].

---

> > ### Author Response · Authors · 2019-11-09
> > **Author Response to Official Blind Review #1, Part 2**
> >
> > >> "Could you clarify which version of PG-MAML was used as the baseline in your experiments?"
> >
> > The PG-MAML we compared with is the vanilla version from Eq. (2) without additional variance reduction. To remain fair in the comparison, we also did not use any variance reduction techniques for vanilla ES-MAML (e.g. we used ForwardFD which has a higher variance than Antithetic). We wanted a comparison between the simplest variants of the two algorithms to understand their fundamental differences.
> >
> > Although applying additional variance reduction techniques as mentioned by the reviewer might improve the performance of PG-MAML, we believe that it will not affect the main conclusions of the paper: fundamentally, PG-MAML still relies on stochastic policy for adaptation, which could lead to catastrophic outcomes in environments for low K and the low relatively low performance for particularly unstable environments shown in Section 4.3, whereas ES-MAML without any particularly complicated variance reduction techniques is able to perform well in these scenarios due to its deterministic policy use and robustness.
> >
> >
> > >> "In section 4.2, with reference to the text “one of the main benefits of ES is due to its ability to train compact linear policies, which can outperform hidden-layer policies”, could you clarify what did this text mean?"
> >
> > We have clarified this point in the paper, thanks.
> >
> > For vanilla ES, linear policies can outperform hidden layer policies [1] (while policy gradient cannot normally train linear policies without modifications such as [2]), and we find that the same setting exists here, where vanilla ES-MAML allows linear policies to outperform hidden layer policies.
> >
> > We make no claims about whether (a necessarily modified variant of) PG-MAML can train with linear policies and whether linear policies under this modified PG-MAML can outperform hidden layer policies under normal PG-MAML. At least a relevant work on this topic is [3], which shows an opposite trend that stacking layers (starting from at least 1 non-linear layer) improves PG-MAML performance, however.
> >
> > >> "1. Page 2, R has not been introduced.
> > 2. Page 3, Section 3.1: does F mean f?"
> >
> > Thanks, we have fixed these typos.
> >
> > [1] Horia Mania Aurelia Guy Benjamin Recht: Simple random search provides a competitive approach to reinforcement learning. NIPS 2018
> >
> > [2] Aravind Rajeswaran, Kendall Lowrey, Emanuel Todorov, Sham Kakade: Towards Generalization and Simplicity in Continuous Control. arXiv:1703.02660, 2018.
> >
> > [3] Chelsea Finn, Sergey Levine. Meta-Learning and Universality: Deep Representations and Gradient Descent can Approximate any Learning Algorithm. arXiv:1710.11622, 2017.
> >
> > [4] Krzysztof Choromanski, Mark Rowland, Vikas Sindhwani, Richard E. Turner, and Adrian Weller. Structured evolution with compact architectures for scalable policy optimization. ICML 2018
> >
> > [5] Yunhao Tang, Krzysztof Choromanski, Alp Kucukelbir. Variance Reduction for Evolution Strategies via Structured Control Variates. arXiv:1906.08868, 2019.

---

### Official Review · AnonReviewer4 · 2019-11-05
**Official Blind Review #4**

**Rating:** 8

**Review:**

The authors propose a new method for model agnostic meta learning (MAML) based on evolution strategies (ES) rather than policy gradients (PG). The proposed method has clear advantages over prior work: it is conceptually much simpler, simpler to implement and is a zero-order method (while PG-MAML requires 2nd order derivatives and differentiation through the update steps).  Also, the method natively allows to incorporate methods from evolution strategies, e.g., to improve exploration. Empirical results are convincing: ES-MAML consistently outperforms PG-MAML (or is at least not worse) on various tasks. Also, ES-MAML seems to be much more robust compared to PG-MAML, which is known to be brittle. The paper is well motivated and well written. The mathematical formalism is precise.


Comment/questions:

- PG-MAML is known to be very sensitive w.r.t. hyperparameters, is this also the case for ES-MAML? How were good hyperparameters found for ES-MAML?
- While this work focuses on RL, it would be interesint to see if ES-MAML is also advantages over vanilla MAML for common few-shot learning image classification problems.
- What’s the efficiency of ES-MAML compared to PG-MAML in terms of wall-clock time?
- (minor:) multiple times in the paper, \citep{} and \citept{} are used incorrectly.



**Experience Assessment:**

I have read many papers in this area.

**Review Assessment: Checking Correctness Of Derivations And Theory:**

I assessed the sensibility of the derivations and theory.

**Review Assessment: Checking Correctness Of Experiments:**

I assessed the sensibility of the experiments.

**Review Assessment: Thoroughness In Paper Reading:**

I made a quick assessment of this paper.

---

> ### Author Response · Authors · 2019-11-09
> **Author Response to Official Blind Review #4, Part 1**
>
> UPDATE(11/12): We posted a new version with some few-shot supervised learning experiments (on sine regression) in Appendix A6 and are still looking at bigger image tasks.
>
> Thank you very much for the encouraging comments!
>
> We provide answers to specific questions below.
>
> >> "PG-MAML is known to be very sensitive w.r.t. hyperparameters, is this also the case for ES-MAML? How were good hyperparameters found for ES-MAML?"
>
> Our implementation of ES-MAML uses two of the techniques from Augmented Random Search [1], an enhancement of the basic Evolution Strategy. Specifically, we use
> - state normalization: record the running mean and standard deviation of the observations in the state space, and normalize the state vectors
> - reward normalization: normalize the reward values used for each estimation of the ES gradient
>
> These enhancements make ES relatively insensitive to the choice of hyperparameters (*), by adjusting the magnitude of the gradients. In fact, we did not perform much tuning, and we used the exact same hyperparameters for different environments and tasks. The inner step size \alpha = 0.05 is inherited from the original PG-MAML paper [2, Appendix A.2]; this was tuned for their PG-MAML experiments but evidently also works for ES-MAML. The other hyperparameters (outer loop step-size, precision parameters) were also fixed across environments and tasks, and were inherited from our defaults on the OpenAI gym benchmarks.
>
> (*) Of course, robustness and reliability remain a challenge in RL [3]. As a comment on the situation, we quote from the paper [1],
> "To put it another way, ARS is not highly sensitive to the choice of hyperparameters because its success rate when varying hyperparameters is similar to its success rate when performing independent trials with a “good” choice of hyperparameters."
>
> >> "While this work focuses on RL, it would be interesting to see if ES-MAML is also advantages over vanilla MAML for common few-shot learning image classification problems."
>
> Yes, we hope to post results in the final version. This requires some additional infrastructure work (in particular Tensorflow) into our distributed ES-MAML code. For supervised learning problems, (subsample) gradients of the loss are available: they can be exactly computed through backpropagation. Thus, for SL, the sensible method is to use backprop for the inner adaptation operator, and ES for the outer loop, leading to a mixed algorithm.
>
> We note an important distinction arises in the supervised setting (SL):
>
> In normal MAML, both SL and RL produce weight parameter updates through a stored buffer of data. For SL, this buffer contains K images from the task dataset, while for RL, this buffer contains K rollout trajectories from the environment.
>
> In the ES case, the total reward is calculated from a trajectory, but since there is no RL replay buffer, the state-action data is thrown away after a trajectory is performed and hence the ES agent is restricted to K queries of the total-reward function for adaptation, in a blackbox fashion.
>
> However, in the SL case, applying this same blackbox-type logic would imply that each adaptation query consists of evaluating the cross-entropy loss on a single new image (which is thrown away afterwards) and there would also be a maximum of K queries allowed. However, the cross-entropy loss on a single image is simply too inaccurate to represent the full dataset’s population loss and this approach would not be sensible. Thus, we opt to instead use a hybrid method, where the inner loop still retains the K images in a buffer and uses a Tensorflow classifier with normal gradient descent initialized from the MAML-point, while the outer loop performs ES optimization.

---

> > ### Author Response · Authors · 2019-11-09
> > **Author Response to Official Blind Review #4, Part 2**
> >
> > >> "What’s the efficiency of ES-MAML compared to PG-MAML in terms of wall-clock time?"
> >
> > This is slightly hard to compare accurately since PG-MAML and First-Order ES-MAML (Appendix A.1) had similarly highly-parallel implementations, whereas Zero-Order ES-MAML was run on fewer cores. The short answer is that PG-MAML took on average 40 seconds for an outer step, and First-Order ES-MAML took about 20 seconds for an outer step. From Figure A.2, Zero- and First-Order ES-MAML are similar in terms of reward/outer iteration, and from Figure 5, PG-MAML and ES-MAML are similar in terms of reward/total rollout, so transitively, similar speeds hold for comparing PG-MAML and Zero-Order ES-MAML.
> >
> > The longer answer is really that "it depends highly on the implementation". Key aspects are:
> > 1. Degree of parallelism - ES can be made highly parallel if desired, see below for details.
> > 2. Distributed computing - we used RPC to parallelize rollouts for ES and the communication time between machines was extremely variable on our network.
> > 3. Policy implementation - PG-MAML used Tensorflow, whereas for ES-MAML we used a lightweight numpy implementation of forward propagation. Empirically, Tensorflow seems to add overhead.
> > 4. Hardware - the machines are heterogeneous, and PG-MAML is more dependent on GPUs.
> >
> > The main cost is the time to evaluate rollouts. Let T denote the time required to execute one rollout, and let K be the number of queries. To evaluate one sample for the outer loop in ES-MAML, it takes T*(K+1) = O(T*K) time if the rollouts are done sequentially, and T + 1 = O(T) if the K queries are done in parallel. Multiple samples (i.e perturbations) can also be parallelized, so if P perturbations are used, then the maximum time if all rollouts is sequential is O(P*T*K), but this is reduced to O(T) if the rollouts for each perturbation are also done in parallel. For PG-MAML, if the meta-policy is running on K parallelized environments, then this also requires O(T) time, and thus is asymptotically similar to ES-MAML.
> >
> >
> > >> "multiple times in the paper, \citep{} and \citept{} are used incorrectly."
> > Thanks! We fixed these issues.
> >
> > [1] Horia Mania, Aurelia Guy, Benjamin Recht: Simple random search provides a competitive approach to reinforcement learning. NIPS 2018
> >
> > [2] Chelsea Finn, Pieter Abbeel, and Sergey Levine. Model-agnostic meta-learning for fast adaptation of deep networks. ICML 2017
> >
> > [3]  Peter Henderson, Riashat Islam, Philip Bachman, Joelle Pineau, Doina Precup, and David Meger. Deep Reinforcement Learning that Matters. AAAI 2018

---

### Official Review · AnonReviewer5 · 2019-11-20
**Official Blind Review #5**

**Rating:** 8

**Review:**

Note: I was asked to write a last-minute review for this paper since the overall rating of the other reviews are not consistent. Therefore, the review is rather brief and I will comment also on concerns raised by the other reviewers.

The paper introduces a new MAML algorithm based on evolutionary strategies (ES) for reinforcement learning tasks. Compared to prior MAML algorithms requiring an estimation of the Hessian, ES-MAML demonstrated to be more stable and efficient. Overall, the paper is well motivated, well written and uses a sound mathematical formulation of the solution approach. Furthermore, the results are convincing and show quite some promise.

Concerning the remarks from Reviewer #3, I believe that it is totally fair to use here a simple ES algorithm that still shows reasonable performance. Of course, we would expect that other ES algorithms might perform better, but this is clearly not the point of the paper. Furthermore, also other papers [1,2] showed that very simple ES algorithm can perform very well on weight optimization of policies.
(Remark: since there is no page limit for refs, I would recommend to cite [1,2] in the paper)

I share some concerns from Reviewer #4 regarding the hyperparameters. By now, it is well known that hyperparameter tuning can improve the performance of RL algorithm quite a bit and is sometimes even the main factor for superior performance. The authors wrote in their reply to Reviewer #4: “In fact, we did not perform much tuning,”. I would like to reply: In fact, this is not a very useful answer.  If there was hyperparameter tuning involved, the amount has to be quantified (in the appendix) and the same amount should be applied to all approaches being compared in the paper.

Furthermore, I missed a discussion about the limitations of the approach. For example, I would expect that the approach will fail if the networks get too large (and thus  the parameter space is too large (>1Mio Parameters?)) and the task is fairly complicated such that the parameter space is not too redundant. I think there is a reason why people tried to use ES for optimizing DNNs for decades, but failed, and now nearly everyone uses GD variants. So, the authors should be more explicit about potential failure cases and limitations.

Small remark: I haven’t found a description of the architectures used in Section 4.4. Since the paper should be self-contained, I would recommend to briefly make this explicit in the appendix.

[1] Patryk Chrabaszcz, Ilya Loshchilov, Frank Hutter: Back to Basics: Benchmarking Canonical Evolution Strategies for Playing Atari. IJCAI 2018: 1419-1426
[2] Lior Fuks, Noor Awad, Frank Hutter, Marius Lindauer:
An Evolution Strategy with Progressive Episode Lengths for Playing Games. IJCAI 2019: 1234-1240

**Experience Assessment:**

I have published one or two papers in this area.

**Review Assessment: Checking Correctness Of Derivations And Theory:**

I assessed the sensibility of the derivations and theory.

**Review Assessment: Checking Correctness Of Experiments:**

I assessed the sensibility of the experiments.

**Review Assessment: Thoroughness In Paper Reading:**

I made a quick assessment of this paper.

---

### Decision · Program_Chairs · 2019-12-19

**Decision:**

Accept (Poster)

**Comment:**

This paper introduces an evolution strategy for solving the MAML problem. Following up on some other evolutionary methods as alternatives for RL algorithms, this ES-MAML algorithm appears to be quite stable and efficient. The idea makes sense, and the experiments appear strong.

The scores of the reviews showed a lot of variance: 1,6,8. Therefore, I asked a 4th reviewer for a tie-breaking review, and he/she gave another 8. The rejecting reviewer mostly took objection to the fact that learning rates / step sizes were not tuned consistently, which can easily change the relative ranking of different ES algorithms. Here, I agree with the authors' rebuttal: the fact that even a simple ES algorithm performs well is very promising, and further tuning would only strengthen that result. Nevertheless, it would be useful to assess the algorithm's sensitivity w.r.t. its learning rate / step size.

In summary, I agree with the tie breaking review and recommend acceptance as a poster.